# Antiviral, Immunomodulatory and Antiproliferative Activities of Recombinant Soluble IFNAR2 without IFN-ß Mediation

**DOI:** 10.3390/jcm9040959

**Published:** 2020-03-31

**Authors:** Isaac Hurtado-Guerrero, Bruno Hernáez, María J. Pinto-Medel, Esther Calonge, José L. Rodriguez-Bada, Patricia Urbaneja, Ana Alonso, Natalia Mena-Vázquez, Pablo Aliaga, Shohreh Issazadeh-Navikas, José Pavia, Laura Leyva, José Alcamí, Antonio Alcamí, Óscar Fernández, Begoña Oliver-Martos

**Affiliations:** 1Instituto de Investigación Biomédica de Málaga-IBIMA, 29009 Málaga, Spain; ishugu@gmail.com (I.H.-G.); mjpmedel@gmail.com (M.J.P.-M.); peperbada@gmail.com (J.L.R.-B.); urbanejaromero@gmail.com (P.U.); anaat73@hotmail.com (A.A.); nataliamenavazquez@gmail.com (N.M.-V.); pabloaliaga.gaspar@gmail.com (P.A.); pavia@uma.es (J.P.); leyvafer@gmail.com (L.L.); oscar.fernandez.sspa@gmail.com (Ó.F.); 2UGC Neurociencias. Hospital Regional Universitario de Málaga, 29010 Málaga, Spain; 3Red Temática de Investigación Cooperativa: Red Española de Esclerosis Múltiple REEM (RD16/0015/0010), 28049 Madrid, Spain; 4Neuroinflammation Unit, Biotech Research and Innovation Centre (BRIC), Faculty of Health and Medical Sciences, Copenhagen Biocentre, University of Copenhagen, 2200 Copenhagen, Denmark; shohreh.issazadeh@bric.ku.dk; 5Centro de Biología Molecular Severo Ochoa, Consejo Superior de Investigaciones Científicas (CSIC)-Universidad Autónoma de Madrid (UAM), 28049 Madrid, Spain; bhernaez@cbm.csic.es (B.H.); aalcami@cbm.csic.es (A.A.); 6AIDS Immunopathology Unit, Centro Nacional de Microbiología, Instituto de Salud Carlos III, Majadahonda 28220 Madrid, Spain; ecalonge@isciii.es (E.C.); ppalcami@isciii.es (J.A.); 7UGC de Reumatología, Hospital Regional Universitario de Málaga, 29009 Málaga, Spain; 8Departamento de Farmacología y Pediatría, Facultad de Medicina. Universidad de Málaga, 29010 Málaga, Spain; 9HIV Unit, Infectious Disease Service, Hospital Universitari de Bellvitge, 08907 Barcelona, Spain

**Keywords:** immunology, soluble receptors, IFNAR, interferon

## Abstract

Soluble receptors of cytokines are able to modify cytokine activities and therefore the immune system, and some have intrinsic biological activities without mediation from their cytokines. The soluble interferon beta (IFN-ß) receptor is generated through alternative splicing of IFNAR2 and has both agonist and antagonist properties for IFN-ß, but its role is unknown. We previously demonstrated that a recombinant human soluble IFN-ß receptor showed intrinsic therapeutic efficacy in a mouse model of multiple sclerosis. Here we evaluate the potential biological activities of recombinant sIFNAR2 without the mediation of IFN-ß in human cells. Recombinant sIFNAR2 down-regulated the production of IL-17 and IFN-ɣ and reduced the cell proliferation rate. Moreover, it showed a strong antiviral activity, fully protecting the cell monolayer after being infected by the virus. Specific inhibitors completely abrogated the antiviral activity of IFN-ß, but not that of the recombinant sIFNAR2, and there was no activation of the JAK-STAT signaling pathway. Consequently, r-sIFNAR2 exerts immunomodulatory, antiproliferative and antiviral activities without IFN-ß mediation, and could be a promising treatment against viral infections and immune-mediated diseases.

## 1. Introduction

Alternative splicing is a process that occurs in approximately 92–94% of human genes [1], whereby a single gene may result in different protein isoforms with different physiological roles [2]. Through this mechanism, many genes that encode cytokine receptors generate soluble isoforms of the receptors, which are released into the bloodstream and represent an important mechanism for modulating the activity of their cytokines. For instance, they can bind to their ligand and act as antagonists by competing with the cell-surface receptor for cytokine binding or they can be agonists, serving as carrier proteins to protect the ligand from proteolysis and improving its stability [3,4,5]. For this reason, these soluble receptors have been used as promising therapeutic targets for the treatment of chronic inflammatory diseases [6].

Interferon beta (IFN-ß) is a cytokine that mediates a variety of biologic responses, including antiviral, antiproliferative, and immunomodulatory effects [7]. Its action is mediated through the interaction with the IFNα/ß cell surface receptor (IFNAR), composed of two subunits (IFNAR1 and IFNAR2), and the activation of the intracellular JAK-STAT signaling pathway [8,9,10]. The IFNAR2 gene can undergo an alternative splicing that generates three isoforms: the short form, which is a non-functional transmembrane protein, the long form, which is composed of the functional receptor together with IFNAR1, and the soluble form (sIFNAR2), which lacks the transmembrane and cytoplasmic domains [11,12] and can be detected in body fluids [13].

In mice, soluble IFNAR2 was proposed as an important regulator of endogenous and systemically administered type I IFN [14]. Other studies have shown that the circulatory half-life of administered IFN-ß could be significantly extended by co-administration with the extracellular domain of IFNAR2 [15]. Besides this, high concentrations of a soluble recombinant form of IFNAR2 neutralized the bioactivity of IFN-ß, while lower concentrations enhanced the IFN-ß-mediated antiviral activity, so the administration of IFN-ß, as a complex with sIFNAR2, may, therefore, enhance the delivery and effectiveness of type I IFNs [16]. A more recent study showed that transgenic mice over-expressing sIFNAR2 are more susceptible to TLR4-mediated septic shock than wild-type mice, by directly exacerbating type I IFN signaling. In concordance with previous studies, sIFNAR2 was also demonstrated to be an important agonist of endogenous IFN actions and is likely to modulate the efficacy of clinically administered IFNs [17].

Given all these antecedents, our initial hypothesis was to potentiate the clinical efficacy of IFN-ß by administrating a recombinant sIFNAR2 cloned in our laboratory. The therapeutic efficacies of the combined treatment (IFN-ß + recombinant sIFNAR2), as well as monotherapy with recombinant sIFNAR2, were evaluated in a chronic mouse model with multiple sclerosis. As a combined therapy, recombinant sIFNAR2 potentiated the immunomodulatory effects of exogenous IFN-ß, but it should be highlighted that its administration as a monotherapy reduced the induced inflammation and tissue damage and exhibited intrinsic antiproliferative activity by inhibiting the T cells’ proliferation [18].

We evaluate whether or not recombinant sIFNAR2 has intrinsic biological activities without the mediation of IFN-ß. New data supporting the antiproliferative and immunomodulatory activity of recombinant sIFNAR2 in human cells are provided, as well as new conclusive data about its antiviral activity, which is independent of the IFN-ß and without mediation of the JAK-STAT signaling pathway. These results uncover a new role for sIFNAR2, providing a potential candidate and strategy for the development of novel immunomodulatory or antiviral therapies.

## 2. Materials and Methods

### 2.1. Cloning of Human Recombinant sIFNAR2

A study of the cloning of the recombinant protein has previously been published [18,19]. Briefly, the prokaryotic expression system pEcoli-Cterm 6xHN Linear (Clontech^®^, Mountain View, CA, USA) was chosen to ligate the sIFNAR2 insert. DH5a Competent Cells™ (Invitrogen^®^, Carlsbad, CA, USA) were transformed with the plasmid and the colony, forming isolated units. After plasmid purification (PureYield™ (Promega^®^, Madison, WI, USA)), BL21 (DE3) (Invitrogen^®^) bacteria were transformed to produce the recombinant sIFNAR2. This was purified on high-capacity Ni^+2^-iminodiacetic acid resin columns, detected by Western Blot using anti-IFNAR2 Human MaxPab antibody (Abnova^®^, Taipei, Taiwan), and its identity was confirmed by the matrix-assisted laser desorption/ionization time of flight (MALDI-TOF) mass spectrometry. The recombinant protein had 239 amino acids and a molecular mass of 29 kDa, and its sequence corresponded to IFNAR2.3 (uniprot: P48551-3). Appendix A shows the identification of the protein by Western blot and by mass spectrometry.

### 2.2. Purification of the Recombinant sIFNAR2 Protein 

Recombinant sIFNAR2 was purified as follows: soluble 6xHN-tagged sIFNAR2 protein from E. coli cell extracts was purified by affinity chromatography (IMAC) using 1 mL HiTrapTM FF crude columnon an ÅKTA FPLC system (GE Healthcare, Chicago, IL, USA), according to the standard procedures. Binding buffer (20 mM sodium phosphate, 0.5 M NaCl, 40 mM imidazole, pH 8.0) and elution buffer (20 mM sodium phosphate, 0.5 M NaCl, 0.5 M imidazole, pH 7.4) were used. Fractions were analyzed in 10% SDS polyacrylamide gels stained with Coomassie Brilliant Blue (BIORAD, Hercules, CA, USA). The sIFNAR2 recombinant protein from different fractions was pooled, dialyzed against PBS and then quantified and loaded on a gel. The final product, with a purity higher than 95%, was stored at −80 °C until used. Protein densitometry was performed using the software ImageJ 1.440 (NIH, Bethesda, MD, USA) to determine its concentration.

The endotoxin levels of the recombinant sIFNAR2 purified were determined using the Endosafe^®^-PTS™ system and the kinetic chromogenic “Limulus Amebocyte Lysate” (LAL) Test Cartridges with a sensitivity of 0.005 EU mL^−1^ (Charles River Laboratories, Wilmington, MA, USA). Endotoxin levels at the working dilutions used for all the experiments were less than 1 EU mL^−1^, which is equivalent to a quantity of endotoxin lower than 0.1 ng mL^−1^ [20,21,22]. Appendix A shows endotoxin levels of recombinant sIFNAR2.

### 2.3. Human Samples

Blood samples from 20 individuals were processed following standard procedures in order to obtain human peripheral blood mononuclear cells (PBMC). Cells were frozen immediately after the reception at the Malaga Regional Hospital Biobank, a node of the Andalusian Public Health System Biobank. All the participants in the study gave their informed consent and protocols were approved by the Institutional Research Ethics Committee (Comisión de Ética de la Investigación Provincial de Málaga. Project identification: DTS 1800045). The project followed the rules of the Declaration of Helsinki (Edinburgh, 2000).

### 2.4. Cell Cultures in the Presence of Recombinant sIFNAR2

PBMC from 19 individuals were thawed and suspended (1 × 10^6^ cells mL^−1^) in pre-warmed RPMI-1640 medium (BioWhittaker, Basel, Switzerland), supplemented with 2 mM l-glutamine (MP Biomedicals, Irvine, CA, USA), 20% Fetal Bovine Serum (FBS) (BioWhittaker, Basel, Switzerland) and 0.032 mg mL^−1^ gentamicin (Normon, Madrid, Spain). Cells were washed by centrifugation and resuspended in RPMI-1640 complete medium with 2% FBS. To evaluate the effect of recombinant sIFNAR2 on the production of secreted cytokines, the time and concentration of the stimulation were established beforehand (4 h, 37 °C and 5% CO_2_) and the cells were stimulated as follows:

Negative control (C−): without stimulation. 

Positive control (C+): PMA (Phorbol myristate acetate, 10 ng mL^−1^) and ionomycin A (1 μg mL^−1^) (Sigma Chemical Co, San Luís, MO, USA).

30 μg mL^−1^ recombinant sIFNAR2 + PMA (10 ng mL^−1^), ionomycin A (1 μg mL^−1^).

To evaluate the effect of recombinant sIFNAR2 on the production of intracellular cytokines, cells from 20 individuals were stimulated in the same conditions as mentioned above, including the protein transport inhibitor (PTI) (BD (GolgiPlug^TM^, 1μL for 10^6^ cells, BD Biosciences, San Jose, CA, USA) in each condition in order to maintain the cytokinesintracellularly.

### 2.5. Determination of Cytokines in Supernatant by Luminex

The multi-analyte detectionof secreted immune mediators (cytokines, chemokines and growth factors) in supernatants was assessed using Procartaplex Multiplex Immunoassay (EPX340-12167-901 ProcartaPlex Human Cytokine and Chemokine Panel 1A) following the manufacturer’s instructions. The plex included the determination of 34 analytes and it was read with the Bioplex 200 platform (Bio-Rad Laboratories Hercules, CA, USA). The results obtained were collected and processed with Bioplex Manager 6.0 software(Bio-Rad laboratories Hercules, CA, USA) using a five-parameter curve-fitting algorithm to analyze the data. Among these 34 analytes, 19 were considered for the analysis because the included controls were optimal, the adjustment of the standard curve was correct, and the intra assay % CV was lower than 5%. The data were represented by a heatmap, within which each analyte was normalized by the maximum value obtained. Each column represents an individual and each row represents an analyte. The greater the intensity of the green color, the greater the expression of the molecule. Further, non-supervised average linkage hierarchical clustering was performed to group the proteins with similar expression patterns. Related sample analyses were performed to compare the production of the secreted immune mediators in the presence of recombinant sIFNAR2, as compared to the positive control.

### 2.6. Determination of Intracellular Cytokines byFlow Cytometry

A dead cell staining kit was used to determine cell viability prior to fixation and the permeabilization required for intracellular antibody staining. In brief, human PBMC were aliquoted (5 × 10^5^ cells/well), washed and stained with monoclonal antibody APC-H7 conjugated anti-CD45 (clone: 2D1; BD Pharmingen, San Diago, CA, USA) and LIVE/DEAD^®^ Fixable Far Red (Invitrogen, Waltham, MA, USA) for 15 min at room temperature and protected from light. After two washes, 100 μL of Fixation/Permeabilization solution (BD biosciences San Jose, CA, USA) was added for 20 min at 4 °C. Cells were then incubated with combinations of the following monoclonal antibodies: PE conjugate anti-IL-17-A (clone SCPL1362) or anti-IL-10 (clone JES3-19F1), and PE-Cy7 anti-IFN-γ (clone B27) or anti-TNFα (clone MAb11) (all from BD Pharmingen, San Diago, CA, USA) for 30 min at 4 °C, in the dark, according to the manufacturer´s instructions. At least 50,000 events from each sample were acquired in a FACSCanto II™ flow cytometer using the FACSDiva software (BD Biosciences, San Jose, CA, USA). Isotype-matched controls were used to verify the staining specificity of the antibodies. The typical forward and side scatter gate of lymphocytes, together with a CD45+ gate, were set to exclude monocytes from the analysis. Related sample analyses were performed to compare the production of the cytokines in the presence of recombinant sIFNAR2, as compared to the positive control.

### 2.7. Analysis of Cell Proliferation

An image-based assay of Cell Proliferation in a live-cell analysis system (IncuCyte S3, ESSEN BioScience, Royston, UK) was carried out. The neuroblastoma cell line Neuro2A (N2A, ATCC CCL-131) was cultured in DMEM containing GlutaMax and supplemented with 10% FBS until they reached 20–40% confluence in the first acquisition. After that, they were exposed to IFN-ß (100 U mL^−1^) or recombinant sIFNAR2 (100 ng mL^−1^) (6 wells per condition) during 40 h (conditions previously established). A non-stimulated control was also included. The IncuCyte was automatically acquired every two hours, 4 images per well, and these images were analyzed, providing data on the percentage of cell confluence (IncuCyte® S3 Live-Cell Analysis System (Cat. No. 4763), ESSEN BioScience, Royston, UK)

### 2.8. Cytopathic Effect Test (CPE) or Bioassay A549/EMCV 

The bioassay is performed as routine test in our laboratory to detect the presence of neutralizing antibodies against IFN-ß (NAbs) in patients treated with IFN-ß for clinical purposes [23,24]. This assay follows the recommendations of “The World Health Organization Expert Committee on Biological Standardization”, which recommends the use of the A549 cell line (human lung cancer cells, ATTC Catalog No. CCL185) and the murine encephalomyocarditis virus (EMCV ATTC Catalog No. VR-129B), so that, when NAbs are present, they block IFN-ß activity and the cell culture is not protected. For the calibration of the A549/EMCV assay, the NIAID (National Institute of Allergy and Infectious Diseases) was requested in order to donate IFN-ß (no. Gb23-902-531) to calibrate the standards of IFN-ß1a and IFN-ß1b. Briefly, 30,000 cells (passage number 85–95) per well were sown on 96 well plates in 100 μL of DMEM, supplemented with 2% FBS, and were incubated at 37 °C and 5% CO_2_ for 24 h. On the second day of the assay, IFN-ß was added to the wells. Each plate included a viral control (VC), which did not contain IFN-ß, a cell control (CC), which did not contain the virus, and a standard IFN-ß (serial dilutions: 1/1 to 1/128). After 24 h of incubation, 100 μL of the EMCV virus was diluted in DMEM medium without FBS (except for the cell control, to which only DMEM without serum was added). In those experiments containing recombinant sIFNAR2 (diluted in DMEM containing 10% FBS), this was added to the culture before and at the same time as infection with the virus. After an additional 24 h of incubation at 37 °C and 5% CO2, the cells were stained for 10 min with crystal violet. The plates were washed to eliminate excess staining and, once dry, the absorbance was read at 630 lambdas in a spectrophotometer. The absorbance values were directly proportional to the cell viability. The absorbance values of each condition were transformed to a percentage of the viability (considering 100% of viability, the mean of the cellular control). In order to demonstrate the intrinsic antiviral activity of recombinant sIFNAR2, bioassays were performed in the presence of different batches of recombinant sIFNAR2, as well as in the presence of albumin (Sigma A4503-BSA), NAbs, B18 (a type I IFN decoy receptor from vaccinia virus, Millipore Cat. # GF156, Burlington, MA, USA) and and Polymyxin B (P4932 Sigma-Aldrich, San Luís, MO, USA).

### 2.9. Determination of p-STAT1 by Flow Cytometry after IFNβ and Recombinant sIFNAR2 Stimulation 

IFN-ß induced the phosphorylation of STAT1 on tyrosine 701, which is a key step in the JAK-STAT signaling pathway. We have previously measured the activation of STAT1 by flow cytometry in human cells in response to IFN-ß, and its relation to the presence of neutralizing antibodies, which block IFN-ß [24]. Thus, the same protocol was followed in order to evaluate the ability of recombinant sIFNAR2 to activate the JAK-STAT signaling pathway. PBMCs, isolated from 5 human subjects, were exposed in duplicate, in two independent experiments, to the following conditions: 

1000 IU mL^−1^ IFNb1a (Avonex, Biogen, Inc.Cambridge, MA, USA).

30 µg mL^−1^ of recombinant sIFNAR2.

RPMI-1640 medium without FBS (unstimulated cells).

Cells were incubated during 30 min at 37 °C to allow for signal transduction and the phosphorylation of STAT1. Following the stimulation, the cells were fixed with Cytofix at 37 °C for 10 min, washed twice with Perm/Wash Buffer, and permeabilized with PermBuffer III (all from BD Biosciences San Jose, CA, USA) at 4 °C for 20 min. The cells were stained for 30 min in the dark, at room temperature, and were labelled with specific monoclonal antibodies for phospho-STAT1 (Y^701^), CD3 and CD8 (all from BD Biosciences San Jose, CA, USA). Isotype-matched controls were used to verify the staining specificity of the antibodies. Cells were acquired in a FACSCanto II™ flow cytometer using the FACSDiva software (BD Biosciences San Jose, CA, USA). At least 50,000 events were acquired from each sample. Unstimulated cells were considered to be at basal levels. The expression of pSTAT1 was determined and analyzed in each sample as a percentage of the positive cells expressing pSTAT1.

### 2.10. MxA Expression by Real Time PCR after IFNβ and Recombinant sIFNAR2 Stimulation

PBMCs, isolated from 5 human subjects, were cultured in the presence of recombinant sIFNAR2 or IFN-ß in order to measure the MxA gene expression, which is an IFN-ß-induced GTP-binding protein. The 8h stimulation time with IFN-ß was previously established as the point in which the MxA expression peaks [18].

PBMCs (1 × 10^6^) were seeded in 12-wells plates, in duplicate, with the following conditions, and in three independent experiments:

Non-stimulated (RPMI-1640 medium without FBS). 

1000 IU mL^−1^ IFNb1a (Avonex). 

30 µg mL^−1^ of recombinant sIFNAR2.

Additionally, an immortalized cell line of human T lymphocytes (Jurkat, Clone E6-1 (ATCC TIB-152)) was used to demonstrate the independence of recombinant sIFNAR2 from IFN-ß, using B18, which blocks IFN-ß activity. Jurkat cells (1 × 10^6^) were seeded in 12-wells plates, in duplicate andwith the following conditions: 

Non stimulated (RPMI-1640 medium without FBS). 

1000 IU mL^−1^ IFNb1a (Avonex). 

1000 IU mL^−1^ IFNb1a (Avonex) + B18 (80 µg mL^−1^).

30 µg mL^−1^ of recombinant sIFNAR2.

To determine the MxA expression, the cells were collected by centrifugation and resuspended in QIAzol Lysis Reagent for RNA extraction (Qiagen, Hilden, Germany). The total RNA was isolated from the cells using a modification of the phenol–chloroform method [25]. The total RNA yield and quality of the product was assessed with the Nanodrop 2000 Spectrophotometer (Thermo Fisher, Waltham, MA, USA). One microgram of the total RNA was reverse transcribed with the M-MLV reverse transcriptase (Sigma) to cDNA, in accordance withthe manufacturer´s instructions. Quantitative PCR was performed in duplicate in a Rotor Gene Q Thermocycler (Qiagen, Hilden, Germany) using RT2 qPCR Primer Assay (Quiagen), KAPA SYBR^®^ FAST Universal One-Step qRT-PCR (Kapa Biosystems, Wildmington, MA, USA) and cDNA. The relative expression of MxA was calculated according to the 2−ΔΔCT method [26], by normalizing to the GAPDH expression.

### 2.11. RNA-Seq and Differential Expression Analyses

Human HeLa cells (ATTC CCL-2) were incubated, or not, with either 50 U mL^−1^ of recombinant human IFNβ (PBL Assay Science, >95% pure) or recombinant sIFNAR2 (30 μg mL^−1^) for 6 h, in duplicate. Cells from each biological duplicate were then harvested and the total cellular RNA was isolated using the SV Total RNA isolation system (Promega, Madison, WI, USA). RNA samples were quantified, and quality analyzed in an Agilent 2100 Bioanalyzer (Agilent Technologies, Santa Clara, CA, USA). All samples exhibited an RNA integrity number (RIN) over 9. The sequencing libraries were generated with the TruSeq RNA Sample Prep Kit v2 Set A (Illumina, San Diego, CA, USA). Briefly, poly(A) containing mRNA molecules were purified in two rounds using oligo-(dT) attached magnetic beads from 1 µg of total RNA. After chemical fragmentation, mRNA fragments were reverse-transcribed and converted into double-stranded cDNA molecules. Following end-repair and dA-tailing, paired-end sequencing adaptors were ligated to the ends of the cDNA fragments using TruSeq PE Cluster Kit v3-cBot-HS (Illumina, San Diego, CA, USA).

Libraries were sequenced using TruSeq SBS Kit v3 - HS (Illumina, San Diego, CA, USA) on an Illumina Hiseq 2500 machine at Centro Nacional de Análisis Genómico (CNAG), Barcelona. More than 4 × 10^7^ 100 nt paired-end reads were obtained from each sample, and after quality assessment with package FastQC (http://www.bioinformatics.babraham.ac.uk/projects/fastqc/), the fastq files containing these reads were mapped to the HeLa genome using Tophat v2.0.4 with default parameters [27].The differential gene expression analysis when comparing with untreated HeLa cells was performed with Cuffdiff (Cufflinks v2.1.0 software, Illumina, San Diego, CA, USA). Pathway analysis of the significantly differentially expressed genes detected was performed using Ingenuity Pathway Analysis (IPA) software. Gene expression heatmaps were generated with the R “Gplots” packages. The raw RNA-Seq data has been deposited at the European Nucleotide Archive (ENA) under the project number (PRJEB33677).

### 2.12. Data Analysis 

Data were analyzed using GraphPad Prism and SPSS 15.0 software. As a non-normal distribution was established in the Kolmogorov–Smirnov test, non-parametric tests were used for comparison between groups. The results are expressed as median and interquartile ranges. Outliers were included in the data analysis and figures. The Wilcoxon Rank test (related samples analysis) was used for the percentage of viability, the cell–culture confluence, the cytokine levels, pSTAT1 level and MxA expression. A Mann–Whitney Utest was used for comparisons related to antiviral activity. Statistical significance was set at *p* < 0.01. The data of the buffer effect, B18, NAbs, albumin and batches were obtained with 8 replicates and were not subjected to statistical analysis. In each experiment, all groups were assigned equal sample sizes. The samples were not randomized, and for all the experiments the untreated and treated samples with recombinant IFNAR2 were run in parallel in order to minimize the inter-assay variation. For the heatmap representation of secreted protein, the data of each metabolite were normalized by the maximum value obtained and were executed with Genesis software [28]. Nineteen metabolites were included as variables in an unsupervised average linkage hierarchical clustering, executed with the same software in order to search for a specific pattern of cytokines in the presence of recombinant sIFNAR2. For antiviral activity data, the absorbance values of each condition were transformed to a percentage of viability, considering 100% of the viability, and the mean of the cellular control.

## 3. Results

### 3.1. Effect of Recombinant sIFNAR2 on the Cytokine Expression Pattern in Human Cells 

The potential immunomodulatory activity of recombinant sIFNAR2 in human PBMC cultures has been evaluated by assessing a panel of secreted cytokines, chemokines and growth factors, as well as the intracellular production of selected pro-inflammatory and anti-inflammatory cytokines in the presence of recombinant sIFNAR2. PMA/ionomycin A was used as a positive control in order to stimulate cytokine production and unstimulated cells were used as negative controls. Appendix A shows PBMC viability in the presence of different recombinant sIFNAR2 concentrations.

#### 3.1.1. Production of Secreted Cytokines, Chemokines and Growth Factors

Figure 1A represents the production of 19 secreted immune mediators in the positive control, negative control, and in the presence of sIFNAR2, as illustrated by the heat maps. As expected, a higher intensity of the green colorrepresents a higher production of the analytes, which is observed in the positive control. However, with the addition of recombinant sIFNAR2, the heatmap shows a lower production of most of the secreted immune mediators analyzed, as compared to the positive control. Through non-supervised average linkage hierarchical clustering, in the presence of recombinant sIFNAR2, different clusters of proteins, according to their production, are identified and differ from those clusters that appear in the positive control. Appendix A shows no induction of IL-17, IFN-γ, TNF-α, IL-4 or IL-10 in human PBMC after recombinant sIFNAR2 stimulation.

Further, the production of GM-CSF, IFN-**γ**, IL-1β, IL-13, IL-18, IL-2, IL-4, IL-6, TNF-α, IL-10, IL-17, Eotaxin, IL-8, IP-10 and MIP-1β, in the presence of recombinant sIFNAR2, were significantly decreased, as compared to the positive control, but showed no differences for IFN-α, IL-1α, MIP-1αand Rantes. Additionally, Figure 1B shows the significant reduction in IL-17, IFN-**γ** and TNF-αproduction in supernatants from cultures treated with sIFNAR2. The medians and interquartile ranges obtained for each immune mediator are summarized in Table 1.

#### 3.1.2. Production of Intracellular Cytokines 

Prior to the cytokine determination procedure by flow cytometry, the cell viability was assessed in order to ensure that it was not altered after recombinant sIFNAR2 exposure. No differences between the percentages of live cells in the positive control, as compared to the percentage of live cells in the presence of recombinant sIFNAR2, were observed (Figure 1D) (more data of cell viability are provided in Appendix A). We then evaluated the intracellular production of IL-17, IFN-γ, TNF-α and IL-10 in the presence of recombinant sIFNAR2 in a lymphocyte gate, based on FSC and SSC properties, as well as CD45 positivity. As expected, the intracellular production of IL-17, IFN-γ, and TNF-α was significantly increased in the positive controlas compared to the negative control, while IL-10 was undetectable in the stimulation conditions tested and was discarded. A significant decrease in the production of IL-17 and IFN-γ was observed in the presence of recombinant sIFNAR2, as compared to the positive control, in concordance with the data observed in the supernatants. However, no differences for TNF-α were observed (Table 1, Figure 1C). 

### 3.2. Antiproliferative Activity of Recombinant sIFNAR2

The antiproliferative activity of recombinant sIFNAR2 has been compared to IFN-ß using the fast-growing mousecell line N2A. The image-based assay showed how the presence of recombinant sIFNAR2 significantly decreased cell confluence over time, when compared to the non-stimulated control, without affecting cell viability. The cell confluence reached in the presence of recombinant sIFNAR2 was similar to that obtained for IFN-ß, whose antiproliferative activity is well-known (Figure 2). These results are in line with our previous data published on mouse T cells [18].

### 3.3. Antiviral Activity of Recombinant sIFNAR2 by CPE (Cytopathic Effect) Inhibition Assay (A549/EMCV)

Thus far, we have demonstrated that recombinant sIFNAR2 is able to exert two out of three biological activities exerted by IFN-ß (immunomodulatory and antiproliferative). As IFN-ß has an important antiviral activity, recombinant sIFNAR2 was tested in the CPE test in order to evaluate this capacity. In these experiments, we tested whether the combination of IFN-ß, plus recombinant sIFNAR2 or the addition of recombinant sIFNAR2 alone, protects the cell culture from the virus infection. 

Independent bioassays were performed to demonstrate the antiviral activity of recombinant sIFNAR2. For each bioassay, serial IFN-ß dilutions were included as a positive control. As can be observed in Figure 3A, the percentage of cell viability in the wells, with dilutions of IFN-ß of up to 1.25 UI mL^−1^, was significantly higher than in the viral control (VC), which means that IFN-ß was able to protect the cells from viral infection in that range of concentrations. Similarly, when three concentrations of recombinant sIFNAR2 (15, 30 and 60 µg mL^−1^) from different batches were added, the percentage of cell viability was always significantly higher than in the VC, demonstrating the intrinsic antiviral ability of recombinant sIFNAR2 to protect the cell monolayer from the virus infection.

Figure 3B is a representative bioassay plate showing the intrinsic antiviral activity of recombinant sIFNAR2. The plate included the CC, VC, IFN-ß standard curve, and different concentrations of IFNAR2. Cells that remained alive after virus infection were stained in blue. In condition 1, serial dilutions of IFN-ß, in combination with a constant concentration of sIFNAR2, were added to the wells; at IFN-ß concentrations below 1.25 UI mL^−1^, in which the protection in the standard curve is lost, the additional presence of recombinant sIFNAR2 was able to maintain cell viability. In conditions 2, 3 and 4, only recombinant sIFNAR2 at three different concentrations was added to the wells, with the cells kept alive, and the cell viability was always higher compared to VC. In condition 5, recombinant sIFNAR2 was added to the cell culture, at the same time that cells were infected with the EMC virus. After 24 h, the absorbance values were similar to the values reached in the CC (where there was no infection), which demonstrates that recombinant sIFNAR2 protects the culture from virus infection and keeps the monolayer intact. The percentage of viability obtained in each condition has been represented in a bar chart.

#### 3.3.1. Experimental Conditions Tested to Demonstrate the Specific Antiviral Activity of the Recombinant sIFNAR2

Elution buffer effect: two dilutions of the elution buffer, without the recombinant protein, were included. The first condition (178 µL buffer + 72 µL DMEM) did not allow for the growing of the monolayer and the wells were almost empty, which was probably due to the lower quantity of growth medium. However, with the working dilution used in all the experiments (107 µL buffer + 143 µL DMEM), the growth of the monolayer was allowed in a normal amount and morphology, but no protection was observed after virus infection and the percentage of viability was similar to the VC (Figure 4A). Additionally, the dilution buffer (PBS 1X) was tested in the bioassay and there was no cell protection (Figure 4H).

Irrelevant protein effect: to rule out the possibility of the antiviral effect being due to the inclusion of a new protein in the media, an irrelevant protein (albumin) was added to the cell culture at the same concentration as sIFNAR2. The cells grew in the presence of albumin, but no protection was observed after virus infection (Figure 4B).

Production/purification process: four different batches of recombinant sIFNAR2, added at their optimal concentration, were tested. Figure 4C shows a complete protection of the cell monolayer with all the batches, where the percentage of viability is always greater in the presence of recombinant sIFNAR2, as compared to the viral control. Figure 4D shows a representative titration curve of batch 8, in which a decrease in the cellular protection, such as the recombinant sIFNAR2 concentration, can be observed, setting the IC_50_ in 33 µg mL^−1^. 

LPS control: in order to discard any possible effect of the residual LPS present in the recombinant sIFNAR2 preparation, Polymyxin B, which is an antibiotic that inhibits LPS, was used. The Figure 4G,I show that in the presence of polymyxin B there is no protection of the cell monolayer. However, the addition of polymyxin B to recombinant sIFNAR2 to block LPS did not affected its antiviral activity, which remained unchanged.

#### 3.3.2. The Antiviral Activity of Recombinant sIFNAR2 is Independent of IFN-ß

In previous experiments, we demonstrated the antiviral activity of recombinant sIFNAR2 independently of the exogenous IFN-ß. In this study, using NAbs and the decoy receptor B18, both of which neutralize the endogenous IFN-ß produced by the cells, we demonstrate that the antiviral effect of recombinant sIFNAR2 is independent, not only of the exogenous IFN-ß, but also of the endogenous IFN-ß.

There was no inhibition of the antiviral activity of recombinant sIFNAR2 in the presence of neutralizing antibodies against IFN-ß (NAbs). A positive human serum, with a high titer of NAbs (917 TRU), was confronted with IFN-ß and sIFNAR2. As expected, in those wells where NAbs blocked IFN-ß, there was no cell protection from the virus infection. However, NAbs did not block sIFNAR2 activity and the monolayer was protected from the virus infection (Figure 4E).

There was no inhibition of the antiviral activity of recombinant sIFNAR2 in the presence of B18. B18, an IFN-binding protein that inhibits IFN-ß antiviral responses, was confronted with IFN-ß and recombinant sIFNAR2. The results showed that the presence of the inhibitor B18 completely abrogated IFN-ß antiviral activity, but had no effect on the antiviral activity of sIFNAR2, whose cells remained completely alive after virus infection (Figure 4F).

Figure 4G shows a bioassay plate including all the effects described above.

### 3.4. The Mechanism of Action of Human Recombinant sIFNAR2 is Independent of the JAK-STAT Signaling Pathway

#### 3.4.1. Recombinant sIFNAR2 Stimulation does not Activate STAT1 in Human Cells

In human PBMC, the activation of the JAK-STAT pathway after stimulation with IFN-ß or recombinant sIFNAR2, was determined measuring he percentage of cells (CD4+ and CD8+ T cells) expressing pSTAT1. As expected, IFN-ß stimulation significantly increased the percentage of cells expressing pSTAT1, as compared to unstimulated cells included as a negative control, in both subpopulations analyzed. However, the stimulation with recombinant sIFNAR2 did not induce the activation of STAT1 and no differences with unstimulated cells were found (Figure 5A), which is in agreement with our previous data on cells isolated from mouse spleens [18].

#### 3.4.2. Recombinant sIFNAR2 Stimulation does not Induce MxA in either Human PBMC or Jurkat Cells

Human PBMC and Jurkat cells were cultured in the presence of recombinant sIFNAR2 to measure MxA expression. As controls, IFN-ß stimulated cells and unstimulated cells were included. In PBMC stimulated with IFN-ß, the MxA expression increased significantly, as compared to unstimulated cells, while there was no induction of MxA in the presence of recombinant sIFNAR2. Similarly, stimulation with IFN-ß increased the MxA expression in Jurkat, as compared to unstimulated cells, but this induction was abrogated in the presence of B18 (Figure 5B,C). Again, stimulation with recombinant sIFNAR2 did not increase the expression of MxA, indicating that recombinant sIFNAR2 did not activate the IFN-ß signaling pathway, which is concordance with our previous observations in mice.

#### 3.4.3. Recombinant sIFNAR2 Elicits a Different Gene Expression Response to IFN-ß

Changes in cellular gene expression after incubation, in the human HeLa cell line with sIFNAR2, appeared using RNA-seq. Among the first 100 pathways, most of which were enriched by those differentially expressed genes (over two-fold change) after sIFNAR2 incubation, we could not detect any pathway related to IFN induction, IFN signaling or IFN effectors. At the same time, we obtained the gene expression pattern from HeLa cells after stimulation with IFN-ß. In this case, we retrieved a set of 96 statistically differentially expressed genes exhibiting a fold change > 2,as compared to untreated cells, including genes with previously demonstrated antiviral activity, such as OASL, IFIT2, MX2, or ISG15, and others recognized as gene regulators of IFN signaling, such as STAT1, STAT2, IRF7 or IRF9. Conversely, when we examined the expression values obtained for this set of genes in those HeLa cells stimulated with sIFNAR2, we could not observe any activation of these IFN induced genes, as compared to untreated cells, reinforcing the fact that sIFNAR2 is not able to elicit signaling via the IFN-ß signaling pathway (Figure 5D). These results confirmed the lack of STAT1 activation observed after sIFNAR2 stimulation of human cells and are concordant with the absence of MxA induction.

After sIFNAR2 stimulation, a large amount of differentially expressed genes was identified. However, the pathways enrichment analysis did not arrive at any obvious conclusion about the sIFNAR2 signaling, since most pathways were redundant and were enriched by a few genes. Among the putative upstream regulators in charge of these changes, the analysis predicted the transmembrane receptor TREM-1, diverse cytokines and growth factors such as TNF, TNFS9IL-1b, PDGF and BMP4 (Appendix A).

## 4. Discussion

An important mechanism to regulate the activity of cytokines, and therefore, the immune system, is the generation of soluble receptors by alternative splicing or proteolysis [3,29,30]. They have a very promising therapeutic potential and some recombinant proteins, corresponding to extracellular portions of cytokine receptors, are used in therapy for inflammatory diseases, autoimmune diseases and other pathologies mediated by cytokines [31]. In the case of IFN-ß, despite all the antecedents describing the soluble isoform of its receptor (sIFNAR2) being capable of modulating the activity of both endogenous and systemically administered IFN-ß [14,16,17], this splice variant has not been studied in depth, even though it was first cloned in 1995 [12].

In the MS context, our starting point was to test the use of the soluble IFN-ß receptor as a coadjutant to IFN-ß in order to improve its activity for clinical purposes; thus, a recombinant extracellular domain of human IFNAR2 was cloned, expressed and purified [19,32]. However, the chronic administration of recombinant sIFNAR2 as monotherapy in the EAE model showed that it was acting as an IFN-ß-like protein, but was independent of IFN-ß. This observation changed our working hypothesis and encouraged us to deepen in its biological activities and its use as a potential drug.

To evaluate its immunomodulatory activity, human cells were cultured in pre-determined conditions in the presence of recombinant sIFNAR2, and the secreted and intracellular production of cytokines and growth factors were measured. As a positive control, the cells were stimulated with PMA, which is a potent activator of protein kinase Cfor ex vivo stimulation and induces the production of cytokines [33]. For the secreted proteins, it can be observed in the heatmap representation how, in the negative control, most of the cytokines and growth factors are present at undetectable levels, and how PMA activates T-cells and enhances their production. However, recombinant sIFNAR2 decreased the production of Th1, Th2 and Th17 cytokines, as compared to the cell activation state achieved with only PMA, because some cytokines are out of the standard curve range of the technique. Consequently, the output of the clustering analysis showed different cytokine expression patterns with recombinant sIFNAR2, as compared to the positive control. Additionally, key cytokine characteristics of Th1 (IFN-γ and TNF-α), Th2 (IL-10) and Th17 (IL-17) profiles were selected for intracellular production. Once we demonstrated that the cell viability was similar in the positive control, and in the presence of recombinant sIFNAR2, we showed that the production of the pro-inflammatory cytokines IL-17 and IFN-γ in CD45+ T cells was diminished in the presence of recombinant sIFNAR2, as compared to the levels achieved with PMA. Consequently, from these two experiments, it can be inferred that recombinant sIFNAR2 decreases the production of both intracellular and secreted pro-inflammatory cytokines IL-17 and IFN-γ, assessed by two independent methodologies. The most pronounced decrease of secreted IL-17, compared to the intracellular IL-17, was probably due to methodological differences.

These effects in the cytokine release are similar to those described for IFN-ß, which has clear anti-inflammatory properties and is well known to prevent Th17 cells from differentiation, reducing the release of IL-17 [34,35] and also to reduce the release of other proinflammatory mediators such IFN-γ [36]. These results point to an immunomodulatory activity of recombinant sIFNAR2 in human cells, in line with previously data observed in the EAE model.

Similarly, its antiproliferative activity was previously described in mouse T cells [18], and now, through a different experimental approach using a time course proliferation assay in a neuroblastoma cell line, recombinant sIFNAR2 decreased the proliferation rate to a similar extent to that observed in the presence of IFN-ß, whose antiproliferative activity is widely known [7].

Regarding the antiviral activity, several bioassays were performed in the presence of 15, 30 and 60 µg mL^−1^ of recombinant sIFNAR2, including appropriate controls. For all the concentrations, the % of cell viability was much higher than in the viral control, the cells remained alive and were stained in blue, which unquestionably proved the antiviral activity of recombinant sIFNAR2. This antiviral activity is similar to that reached in the presence of IFN-ß, which has a potent antiviral activity. In a representative example, e.g., Figure 3B, the monotherapy with different concentrations of sIFNAR2 kept the cell monolayer intact after infection with the virus. In addition, in combination with lower IFN-ß concentrations, which no longer protected the cells, the cell viability observed is due to the presence of recombinant sIFNAR2. However, in combination with higher IFN-ß concentrations, no further synergistic effects were observed.

To ensure that the protection of the cell culture against the viral infection was due to the presence of recombinant sIFNAR2, the buffer effect and the use of an irrelevant protein was analyzed. The lower dilution of the buffer did not have the appropriate growth composition (nutrients, pH and osmolality) to maintain the monolayer. However, with the higher dilution tested, the monolayer grew normally and remained intact, but no protection was observed after viral infection, discarding the buffer effect. Additionally, using albumin at the same working concentration as the recombinant sIFNAR2, there was also no protection. Both bioassays demonstrated that there is no protection in the absence of recombinant sIFNAR2.

In addition, the antiviral activity of recombinant sIFNAR2 has been shown to be independent of the de-production/purification process of the protein, and all the tested batches exerted this activity. While there is an inherent variation in the purification process of recombinant proteins, all of our batches have been able to protect the cells from virus infection in the range from 15 to 60 µg mL^−1^. 

Furthermore and most interestingly, our data demonstrate that the antiviral activity of recombinant sIFNAR2 is independent of IFN-ß because it was not inhibited by the presence of the decoy receptor B18, nor at high doses of NAbs, while both did inhibit the antiviral activity of IFN-ß, which was not able to protect the cell monolayer.

One of the biggest drawbacks of the recombinant proteins produced in *E.coli* is the contamination with endotoxins, where the major component is lipopolysaccharide [37]. It has been shown that even residual endotoxin contaminations are able to stimulate monocytes via TLR4/ CD14 (CD14 Residual Endotoxin [38]). Along the same lines, one limitation of our study could be the putative influence of LPS on the final effect of sIFNAR2. In our hands, the effects of the potential residual endotoxin contamination have been very low or negligible as shown in results. Even in PBMC, particularly sensitive to LPS, we have not found production of cytokines classically induced by LPS like IFN-γ or TNF-α. For the antiviral activity, we have used polymyxin B to block the residual endotoxin contamination and the antiviral activity of sIFNAR2 remained unchanged. In addition, the RNA-seq expression pattern of sIFNAR2 modified a large number of genes and only 20 are overlapping with the LPS signature.

In 2001, Han et al. described a recombinant ovine soluble form of IFNAR2 and IFNAR1 with antiviral activity in the absence of IFN [39], which strongly supports our results concerning the intrinsic antiviral activity of human sIFNAR2, as described herein for the first time. While this antiviral activity has been overlooked for a long time for the soluble IFN-ß receptor, it has been described recently for the soluble IL-6 receptor (sIL6R) [40]. IL-6, similarly to IFN-ß, is a cytokine involved in the acute inflammatory response to viral infection [41] and the sIL6R exhibited an extensive antiviral activity against DNA and RNA viruses without IL-6 mediation. Here we demonstrate the antiviral activity of recombinant sIFNAR2 without IFN-ß mediation.

The secretion of soluble proteins that bind type I IFN with high affinities and prevent its interaction with IFNAR is an efficient strategy of several viruses to counteract the IFN antiviral activity in host responses [42,43,44]. However, the demonstration of the soluble forms of cytokine receptors, like sIFNAR2 or sIL6R, which are biologically active without the participation of the cytokine, suggest that they could have a more relevant role in the immune system, but little is known about them.

Regarding the signaling events, previous studies described the inhibitory or enhancing action of sIFNAR2 as being mediated through the activation of the classical IFN-ß signaling pathway (JAK-STAT) [14,16,17]. However, we demonstrated, in murine spleen-T cells, that our recombinant sIFNAR2 was not able to activate the JAK-STAT signaling pathway per se because there was no activation of IFN-ß-inducible proteins such as pSTAT1 or MxA [18]. While its signaling pathway is not elucidated yet, our new results reinforce, once more, the independence of the sIFNAR2 from IFN-ß and from the JAK-STAT pathway.

As the IFN-ß response has been demonstrated to be specific to cell types [45], the activation of the JAK-STAT signaling, in the presence of recombinant sIFNAR2, has been evaluated in human cells. There was no phosphorylation of STAT1 in sIFNAR2 stimulated human T cells, either through the induction of MXA in human PBMC or Jurkat cells. Later, in order to explore the potentiallyenriched pathways, an RNA-seq was performed in HeLa cells after treatment with either IFN-ß or recombinant sIFNAR2. The IFN-ß treatment induced the differential expression of 96 genes that are classically induced by IFN-ß [46]. However, for this set of genes, we could not observe any induction in the presence of recombinant sIFNAR2. RNA-seq results are concordant with the absence of MxA induction in Jurkat and PBMC, as assessed by PCR, and the lack of activation of STAT1 at protein levels. These results indicate that recombinant sIFNAR2 is not able to elicit a signaling response that is similar to IFN-ß. Furthermore, the pathways enrichment analysis after sIFNAR2 stimulation did not allow for a conclusion concerning its signalling, since most pathways were redundant and enriched by only a few genes. Further RNA-seq experiments are necessary, with different induction times to identify alternative specific cellular pathways triggered by sIFNAR2.

## 5. Conclusions

The present work demonstrates that recombinant sIFNAR2 has intrinsic biological activities without the mediation of IFN-ß: 1. it reduces the pro-inflammatory cytokines profile in human cells; 2. it has an intrinsic antiviral and antiproliferative activity; and 3, its activities are independent of the JAK-STAT signaling pathway. Altogether, these results suggest that recombinant sIFNAR2 is a promising candidate for treatment against viral infections and immune-mediated diseases.

## 6. Patents

I.H.-G, M.J.P.-M L.L., J.P., O.F., and B.O.-M are inventors of a family of patents covering recombinant sIFNAR2.

## Figures and Tables

**Figure 1 jcm-09-00959-f001:**
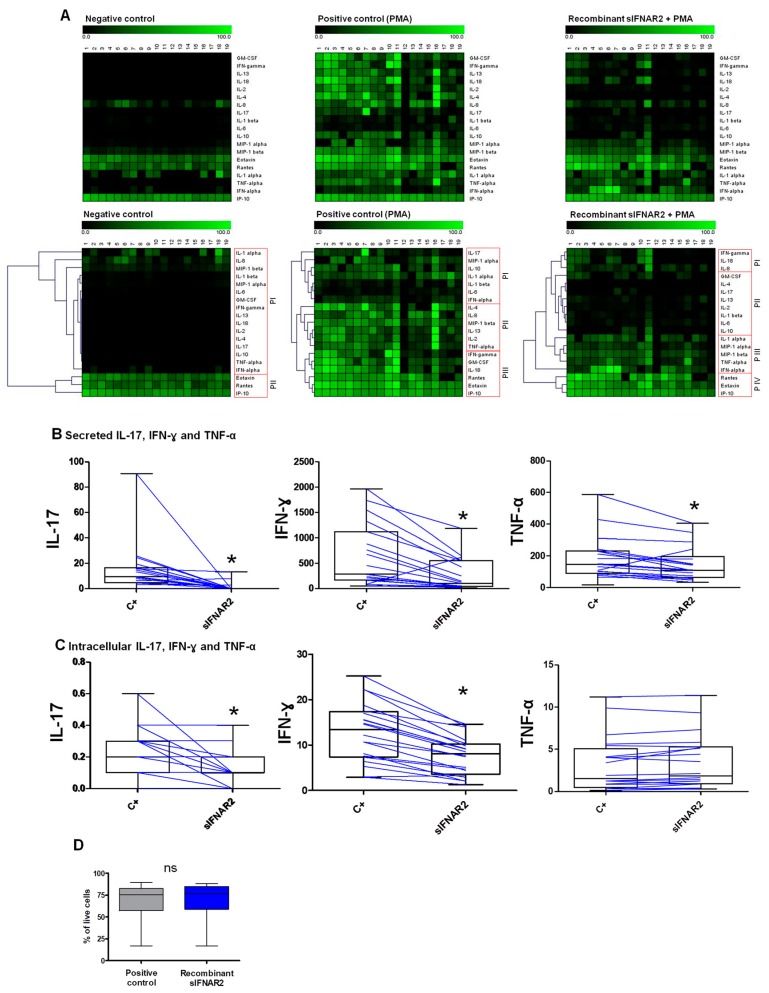
Immunomodulatory activity of recombinant sIFNAR2. (**A**) Heatmaps showing the production of secreted cytokines, chemokines and growth factors in human cells in the presence of recombinant sIFNAR2. Each metabolite was normalized by the maximum value obtained. Each column represents a subject and each row represents a metabolite. A higher intensity of the green color represents a higher production of the molecule (above). Non-supervised average linkage hierarchical clustering grouped the proteins with similar expression patterns (P) and they were numbered PI, PII, PIII or PIV in each map. The clustering was executed with Genesis software (below). (**B**) Secreted IL-17, IFN-ɣ, and TNF-ɑ were assessed by Luminex (pg mL^−1^).The data have been depicted in groups (as box-plots) and individually (as lines in order to show intra-individual changes). The cells exposed to recombinant sIFNAR2 were compared to the positive control (c+) (Wilcoxon Rank test). *N* =19 * *p*< 0.01. (**C**) Intracellular IL-17, IFN-ɣ, and TNF-ɑ in CD45+ T cells were assessed by flow cytometry. The data show the percentage of positive cells expressing IL-17, IFN-ɣ, and TNF-ɑ. The data have been depicted in groups (as box-plots) and individually (as lines in order to show intra-individual changes). The cells exposed to recombinant sIFNAR2 were compared to the positive control (c+) (Wilcoxon Rank test). *N* = 20 * *p*< 0.01. (**D**) Cell viability before the fixation and permeabilization of cells for the evaluation of the production of intracellular cytokines by flow cytometry using LIVE/DEAD^®^ Fixable Far Red. *N* = 20. ns: non-significant.

**Figure 2 jcm-09-00959-f002:**
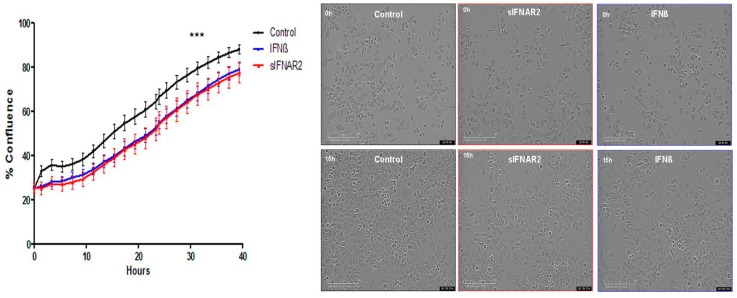
Antiproliferative activity of recombinant sIFNAR2. The graph shows the mean percentage ± s.d. confluence of the N2a cell culture over time. The % of confluence in the presence of recombinant sIFNAR2 was compared to that of the control (Wilcoxon Rank test). *** *p*< 0.01. Representative images of the cell culture before (above) and after 15 h of treatment (below).

**Figure 3 jcm-09-00959-f003:**
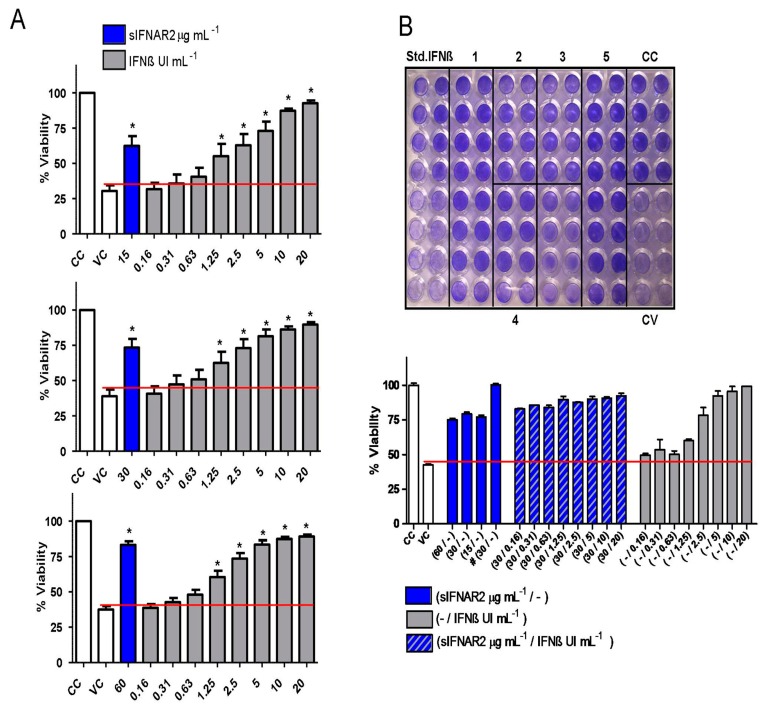
Antiviral activity of recombinant sIFNAR2. (**A**) Demonstration of the antiviral activity of recombinant sIFNAR2 by the cytopathic effect test (CPE) or bioassay. The bar chart represents the mean of the percentage of viability ± s.d. in the presence of 15 30 y 60 µg mL^−1^ of recombinant sIFNAR2 (*N* = 10 for 15 and 30 µg mL^−1^ and *N* = 18 for 60 µg mL^−1^), including the viral control (VC), the cellular control (CC) and the IFN-ß standard curve (from 20 to 0.16 UI mL^−1^). Values above the red line indicate cell protection after virus infection. The % of viability, in the presence of the different concentrations of recombinant sIFNAR2 or IFN-ß, was compared to the % of viability of the virus control (Mann Whitney test). * *p* < 0.01. (**B**) Image of a representative CPE plate performed with a A549 cell line and infected with the EMC virus (above). Non-infected cells (CC), as well as those cells that remained alive after the virus infection, were stained in blue. The assay included the following conditions: IFN-ß standard curve (from 20 to 0.16 UI mL^−1^); (1) decreased concentrations of IFN-ß (from 20 to 0.16 UI mL^−1^) + constant recombinant sIFNAR2 (30 µg mL^−1^); (2, 3 and 4) eight replicates of 15, 30 and 60 µg mL^−1^, respectively, of recombinant sIFNAR2 added the day before the infection; (5) sixteen replicates with 30 µg mL^−1^ of recombinant sIFNAR2 added at the same time as the virus (#); CC: cellular control; and VC: viral control. Below, the bar chart represents the mean of the percentage of viability ± s.d. of each condition from the CPE plate.

**Figure 4 jcm-09-00959-f004:**
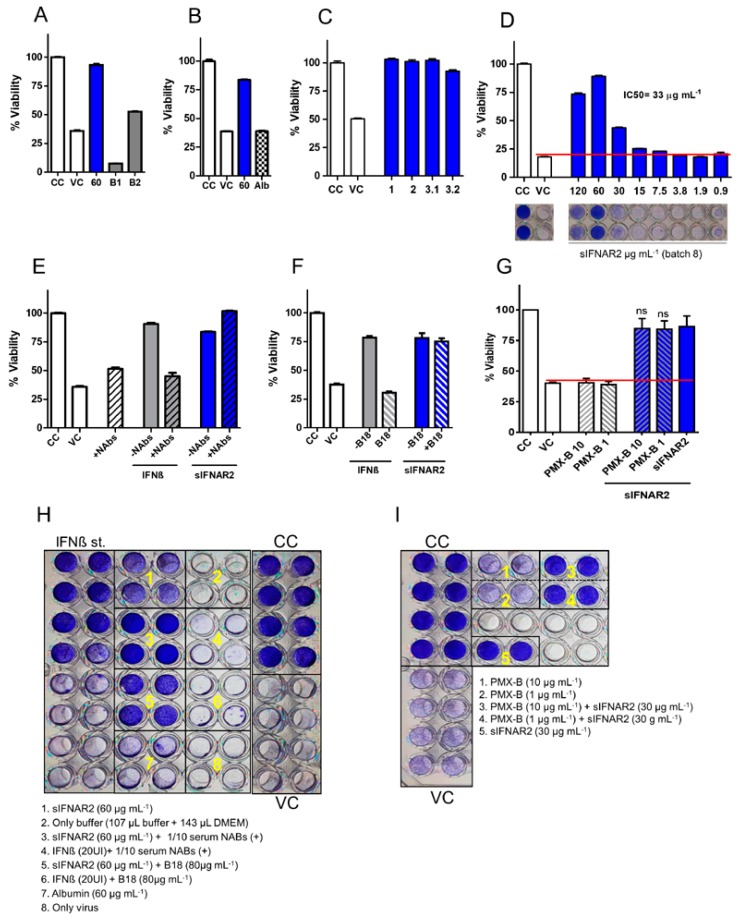
Specificity of the antiviral activity of the recombinant sIFNAR2. (**A**) Buffer effect. The CPE test was performed with two dilutions of the buffer (B1 and B2) without recombinant sIFNAR2. The B1 did not allow for the growth of the monolayer and the wells were almost empty due to the lower quantity of DMEM. The B2 allowed for the growth of the monolayer, but showed no protection after virus infection. (**B**) Irrelevant protein. The CPE test was performed in the presence of 30 µg mL^−1^ of albumin and no protection was observed after virus infection. For both graphs, the bar chart represents the mean of the percentage of viability ± s.d. of 8 duplicates in each condition. (**C**) Production/purification process. The CPE test was performed with four different batches of recombinant sIFNAR2 (1, 2, 3.1 and 3.2) at 30 µg mL^−1^. The bar chart represents the mean of the percentage of viability ± s.d. of 8 duplicates. (**D**) Representative dose-response curve of recombinant sIFNAR2 (from 120 to 0.9 µg mL^−1^) in duplicate. 120 µg mL^−1^ did not allow for the confluence of the monolayer, and the maximum antiviral effect was achieved at 60 µg mL^−1^. The IC50 was established at 33 µg mL^−1^. (**E**) Neutralizing antibodies against IFN-ß effects (NAbs). The CPE test was performed in the presence of a high titer of NAbs (917 TRU) that blocked the IFN-ß antiviral activity, but not the recombinant sIFNAR2 (at 60 µg mL^−1^) antiviral activity. The bar chart represents the mean of the percentage of viability ± s.d. of each condition (NAbs / IFN-ß UI mL^−1^ or sIFNAR2 µg mL^−1^) of 8 replicates. -NAbs: serum without neutralizing antibodies. +NAbs: serum with neutralizing antibodies. (**F**) Presence of IFN-ß inhibitor (B18). The CPE test was performed in the presence of B18, which abrogated the IFN-ß antiviral activity, but not the recombinant sIFNAR2 antiviral activity. The bar chart represents the mean ± s.d of the percentage of viability of each condition in 8 replicates.(**G**)Bioassays were performed in the presence of polymyxin B (PMX-B) in order to block the residual LPS of the recombinant sIFNAR2 preparation. Polymyxin B was used at 10 and 1 µg mL^−1^ alone or in the presence of recombinant sIFNAR2 (at 30 µg mL^−1^). The presence of polymyxin B did not affect the antiviral activity of recombinant sIFNAR2. The bar chart represents the mean ± s.d of the percentage of viability of each condition for three independent experiments. CC: cellular control. VC: viral control. (**H**)Bioassay plate showing the effect of the buffer and of an irrelevant protein (albumin), as well as how NABs and B18 inhibit the antiviral activity of IFN-ß, but not the antiviral activity of recombinant sIFNAR2. (**I**) Bioassay plate showing that the presence of polymyxin Bdid not affect the antiviral activity of recombinant sIFNAR2. CC: cellular control. VC: viral control.

**Figure 5 jcm-09-00959-f005:**
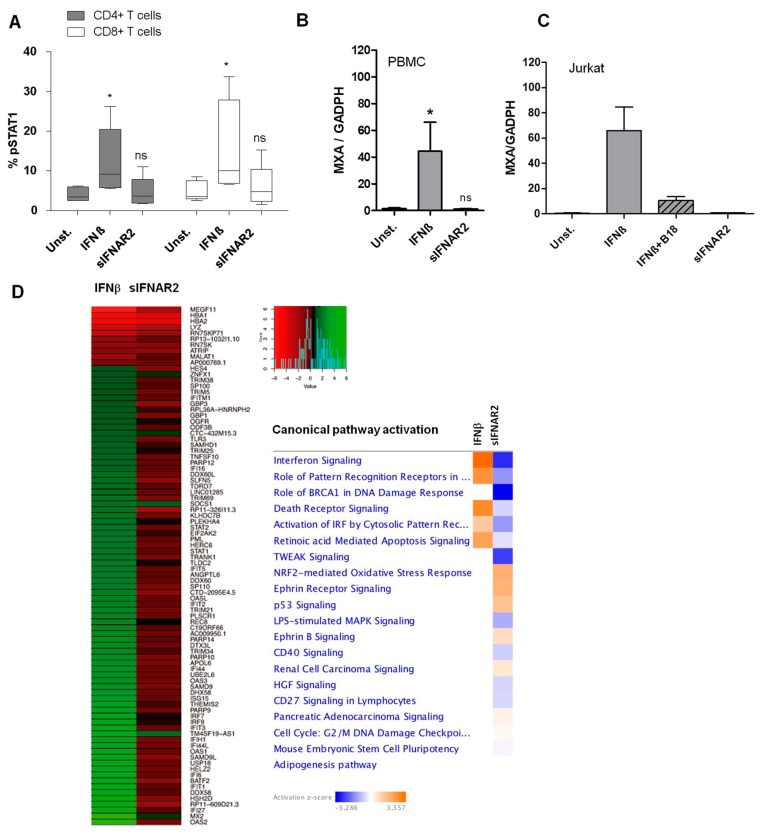
The activities of human recombinant sIFNAR2 are independent of the JAK-STAT signaling pathway. (**A**) Percentage of pSTAT1 in unstimulated (unst.) CD4+ and CD8+ human T cells and after stimulation with IFNβ or with recombinant sIFNAR2, assessed by flow cytometry (*N* = 5). sIFNAR2 does not induce STAT1 phosphorylation (Wilcoxon Rank test). * *p*< 0.01, ns. was not significant. (**B**) Relative MxA gene expression in PBMC. MxA expression in unstimulated PBMCs was compared to that from PBMC stimulated with IFN-ß or sIFNAR2. (Wilcoxon Rank test). **p*< 0.01, ns. not significant. sIFNAR2 did not induce MxA expression. (**C**) Relative MxA gene expression in the Jurkat T-cell line, including inhibitor B18. Stimulation with IFN-ß increases the MxA expression, as compared to unstimulated cells, and this induction was abrogated in the presence of B18. sIFNAR2 did not induce MxA expression. (**D**) Differential gene expression and canonical pathway activation analyses in the presence of recombinant sIFNAR2. Gene expression was determined by RNA-seq from HeLa cells before and after stimulation with either IFN-ß or sIFNAR2. The heatmap displays the 96 statistically differentially expressed genes (fold change > 2 and *p* value < 0.05) identified after the comparison to untreated cells (Left). Enriched canonical pathways after IPA analysis with the corresponding statistically differentially expressed genes determined in each case. Color indicates the pathway induction (orange) or inactivation (blue) (Right).

**Table 1 jcm-09-00959-t001:** Levels of secreted cytokines and intracellular cytokines.

Secreted Cytokines	Positive Control	Recombinant sIFNAR2	P
GM-CSF	74.29 (36.41–123.41)	0 (0–30.1)	<0.01
IFN-γ	286.41 (160.79–1119.49)	100.91 (36.66–547.09)	<0.01
IL-1β	3.68 (2.28–6.33)	0.52 (0–2.89)	<0.01
IL-13	28.62 (16.95–43.87)	4.98 (3.31–10.84)	<0.01
IL-18	19.05 (9.27–39.25)	0 (0–15.79)	<0.01
IL-2	407.85 (226.78–751.98)	70.33 (31.82–92.4)	<0.01
IL-4	23.24 (12.48–41.05)	0 (0–0)	<0.01
IL-6	181.26 (124.48–280.64)	126.23 (32.47–204.91)	<0.01
TNF-α	146.03 (87.17–230.89)	107 (59.46–179.44)	<0.01
IL-10	4.33 (2.68–5.81)	0 (0–0)	<0.01
IL-17	9.46 (4.28–16.37)	0 (0–0)	<0.01
IFN-α	0.55 (0–1.01)	1.25 (0.23–2.81)	ns
IL-1α	0.65 (0–1.09)	0.6 (−0.81–1.15)	ns
Eotaxin	1.66 (1.06–1.97)	0.67 (0.03–0.89)	<0.01
IL-8	1632.76 (899.79–3879.95)	192.61 (−757.7–553.58)	<0.01
IP-10	1632.61 (900.96–3877.33)	191.59 (−757.85–552.85)	<0.01
MIP-1α	154.84 (−12.54–313.45)	118.97 (59.33–187.69)	ns
MIP-1β	730.09 (408.43–1047.2)	432.29 (300.74–528.52)	<0.01
Rantes	21.73 (16.37–55.69)	49.1 (−7.98–90.23)	ns
**Intracellular Cytokines**	**Positive Control**	**Recombinant sIFNAR2**	**P**
IL-17	0.2 (0.2–0.3)	0.1(0.1–0.2)	<0.01
IFN-γ	13.4 (7.3–17.37)	8.1 (3.57–10.25)	<0.01
TNF-α	1.5 (0.42–0.07)	1.8 (0.9–5.2)	ns

Levels of normalized secreted cytokines (pg mL^−1^) and intra-cellular cytokines expressed as a % of cell expressing the cytokine. Data are medians and interquartile ranges. The Wilcoxon Rank test was used to compare the effect of recombinant sIFNAR2 with the positive control.

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
