# Peer review of "Antiviral, Immunomodulatory and Antiproliferative Activities of Recombinant Soluble IFNAR2 without IFN-ß Mediation"

_jcm, 2020, doi:10.3390/jcm9040959_

Round 1
Reviewer 1 Report
This manuscript reports an original finding, namely an unexpected “IFN-like” immunomodulatory, antiproliferative and antiviral activity of sIFNAR2 in vitro in human cells, which might represent a significant advance in the IFN field. However, since sINFAR2 was produced in bacteria, a possible contamination with endotoxin/LPS might explain, at least partly, some (but not all) of the biological results observed. Unfortunately, the new version of the manuscript raises more questions than it solves. First, cytokine production in primary PBMCs is highly sensitive to even minor LPS contamination. Either additional control experiments need to be performed or the results in PBMCs need to be removed. Second, cell lines such as A549 or HeLa are less sensitive to LPS but can still respond to it. In addition, very low levels of type I IFN induced by the traces of LPS contamination will be neutralized by sIFNAR2, while other pro-inflammatory mediators are not, complicating the overall biological interpretation of each experiment, as these effects will differ in each line.
Major comments:
- The authors did perform the requested LPS quantification using 2 different methods, which actually revealed detectable LPS contamination of the major reagent of this manuscript, sIFNAR2. However, the authors do not mention the outcome of these measures, but indirectly report that “Endotoxin levels at the working dilutions used for all the experiments were less than 1 EU ml-1, which is equivalent to a quantity of endotoxin lower than 0.1 ng ml-1 [20–22].” The authors should report the absolute concentration of endotoxin by the two methods as EU/mg protein or ng/mg protein, in the four different batches that were used in the manuscript. Moreover, 0.1 ng/ml or 100 pg/ml of LPS already provides a strong stimulus of PBMCs or primary human monocytes and is definitely not negligible. This casts strong doubts on the results obtained with PBMCs in this manuscript, especially since the control condition of sIFNAR2 without PMA is missing in Figure 1. Were these data left out because sIFNAR2 by itself induces cytokine production?
- Moreover, the authors did NOT perform the requested control experiment with polymyxin B in PBMCs, to measure the effect of LPS contamination on the pro- or anti-inflammatory cytokine induction under sIFNAR2 stimulation.
- The authors did perform a control experiment with polymyxin B, not in PBMCs (as requested) but in a cell line, A549, which has much lower responsiveness to LPS. In addition, the authors did not preincubate (as requested, = correct lab practice for LPS contamination) the concentrated sIFNAR2 with polymyxin B before adding it to the cell cultures, which is absolutely necessary to neutralize endotoxin before it can bind to its cellular receptor(s). In addition, Figure 4H shows a single experiment with polymyxin B instead of a mean/median of three experiments with statistics, whereas the picture of the wells with sIFNAR2+polymyxin B suggest a (small) inhibition when comparing wells in 3-4-5.
- Therefore, the entire paragraph on LPS in the Discussion is incorrect: “In addition to the low level of endotoxins, which contain our preparation, we have important evidence that supports the fact that the observed effects with recombinant sIFNAR2 are not mediated by LPS, such as: 1. decrease or no induction of cytokines classically induced by LPS. 2. Protection of the cell monolayer after virus infection in the presence of an LPS inhibitor 3. The RNA-seq expression pattern of sIFNAR2 does not match the LPS RNA-seq expression pattern.”
4.1. As discussed above in 1., the control of sIFNAR2 without PMA is lacking for cytokine production in PBMCs.
4.2. As discussed in 3., statistics from several experiments are needed to make this claim.
4.3. An LPS gene signature is enriched in the authors’ RNAseq data by GSEA (MSygDB): NT (non-tolerizeable) genes induced during the first LPS stimulation and induced at equal or greater degree in tolerant macrophages (20 genes overlapping,enrichment p=5.4x10-7). These genes include cytokines CSF-1, TNFSF4 and TNFSF9, all of which are LPS-inducible and overexpressed in sIFNAR2-treated HeLa cells.
Minor comments:
- Although a minor revision request, the authors did not do a basic English grammar revision. Several errors remain (or have been introduced) in the new version of the manuscript. For instance, line 457-459: “In order to discard any possible effect of the residual LPS presents in the recombinant sIFNAR2 preparation”: present not presents; “Polymyxin B, that is an antibiotic that inhibits LPS”: which is an antibiotic; “However, the addition of polymyxin B to recombinant sIFNAR2 for blocking LPS, did not affected its antiviral activity, that remained unchanged.”: “to block LPS, did not affect”.
Line 474-475 “a IFN-binding protein that inhibit IFNß antiviral responses” instead of “an IFN-binding protein”
Author Response
Enclosed you can find the reviewer response.

Reviewer 2 Report
The authors have addressed all my previous comments.
Author Response
Thank you so much
Begoña
Round 2
Reviewer 1 Report
I agree with all changes made by the authors to the current manuscript and I fully understand the requested experiment with primary PBMCs is not possible in the near future due to the consequences of the COVID-19 outbreak.
However, I would like to repeat my request to please check the nomenclature of cytokines, as this manuscript will appear with ‘Immunology’ as a keyword. As I mentioned before under ‘Minor comments’, point 2: “Please use correct names for cytokines and use consistently throughout the manuscript (text and figures): GM-CSF (not GM-CFS), IFN-alpha or IFN-α (not IFNa), idem for IFN-gamma, TNF-alpha, also IL-10 (not IL10) etc.”
While I admit that an occasional hyphen might slip in any cytokine paper, such that IL-2 appears as IL2, or that the use of “IFN-gamma” will not confound the readers (although IFN-ɣ is the official nomenclature), the current manuscript still has numerous mistakes. For instance, there are twelve (!) mentions of “TNF-ɣ”, even in the abstract, which is just not acceptable.
Author Response
I agree with all changes made by the authors to the current manuscript and I fully understand the requested experiment with primary PBMCs is not possible in the near future due to the consequences of the COVID-19 outbreak.
Thank you so much for understanding the situation.
However, I would like to repeat my request to please check the nomenclature of cytokines, as this manuscript will appear with ‘Immunology’ as a keyword. As I mentioned before under ‘Minor comments’, point 2: “Please use correct names for cytokines and use consistently throughout the manuscript (text and figures): GM-CSF (not GM-CFS), IFN-alpha or IFN-α (not IFNa), idem for IFN-gamma, TNF-alpha, also IL-10 (not IL10) etc.”
While I admit that an occasional hyphen might slip in any cytokine paper, such that IL-2 appears as IL2, or that the use of “IFN-gamma” will not confound the readers (although IFN-ɣ is the official nomenclature), the current manuscript still has numerous mistakes. For instance, there are twelve (!) mentions of “TNF-ɣ”, even in the abstract, which is just not acceptable.
All the nomenclature of cytokines has been reviewed as better as possible
Thank you so much for your effort reviewing our work.
Begoña
This manuscript is a resubmission of an earlier submission. The following is a list of the peer review reports and author responses from that submission.
Round 1
Reviewer 1 Report
In the current manuscript, entitled “Antiviral, immunomodulatory and antiproliferative activities of recombinant soluble IFNAR2 without IFNß mediation”, Hurtado-Guerrero and coworkers show extensive and potent “IFN-like” bioactivity of the soluble IFNAR2 receptor in the absence of its ligand. The experiments provide convincing evidence for antiviral, immunomodulatory and antiproliferative activity of sIFNAR2, which is of strong clinical relevance. However, an additional control experiment to rule out possible LPS contamination of the recombinant protein should be performed. In addition, more experimental detail should be provided, on the source and purity of albumin and B18 Vaccinia receptor, and particularly on the systems biology analysis.
Major comments:
Although another protein was tested (albumin), no details on its purification or source (bovine or human serum albumin?) is provided. In contrast, sIFNAR2 is produced in E. coli, which carries the risk for LPS contamination, even in trace amounts. Since LPS is a common contaminant in recombinant proteins and strongly induces cytokine production at sub-nanomolar levels, the authors should check blocking LPS with polymyxin B. Detection of LPS by Limulus assays is a good start for strong LPS contamination but is not sufficient for “mild” LPS contamination, since the detection limit of the assay (0.01 EU/ml) corresponds to +/- 100 pg/ml of coli endotoxin, while 1-10 pg/ml of LPS already induces strong cytokine production by human PBMCs or monocytes. A simple control experiment using sIFNAR2 vs. LPS in the absence or presence of polymyxin B (incubating the concentrated protein or LPS with polymyxin B before adding to the PBMC cultures) in PBMCs from a few (3-6) healthy donors would be sufficient to check this. Since sIFNAR2 appears to exert an IFN-like activity in the absence of STAT1 activation, the RNA-seq experiment and systems biology analysis are extremely relevant to shed light on the signaling pathways triggered by sIFNAR2. However, the authors remain vague and do not suggest any pathways or networks, although IPA analysis was performed. Therefore, a complete list of differentially expressed genes should be provided (with raw p-values as well as FDR-corrected) along with the significant pathways, networks and upstream regulators by IPA (as supplementary material). Given the wealth of data IPA analysis provides (at least if sufficient genes are differentially expressed), the authors should use this to at least suggest some signaling pathways triggered by sIFNAR2 to exert the large spectrum of biological activity observed in this manuscript.
Minor comments:
Language and style need to be checked for the entire manuscript: some phrases are difficult to follow and unnecessary long e.g. “To evaluate which of these changes reached statistical significance, related samples analysis WAS performed to compare the production of the secreted immune mediators in presence of recombinant sIFNAR2 compared to the positive control.” Related samples analysis WAS performed to compare the production of the cytokines in THE presence of recombinant sIFNAR2 compared to the positive control.” The authors often use vague or imprecise wording, which makes it difficult for the reader to understand what they actually mean. In line 573, why was there insufficient medium in the first place? “The lower dilution of buffer tested did not allow the growth of the monolayer because of the insufficient quantity of medium.”
Several phrases lack concordance (e.g. the cell line… they) and overall the absence or presence of definite articles is erratic. “The neuroblastoma cell line Neuro2A (N2A, ATCC CCL-131) was cultured in DMEM 165 containing GlutaMax supplemented with 10% FBS until IT reachED 20-40% confluence in the first acquisition. The potential immunomodulatory activity of recombinant sIFNAR2 in human PBMC cultures has been evaluated by assessing a panel of secreted cytokines, chemokines and growth factors, as well as the intracellular production of selected pro-inflammatory and anti-inflammatory cytokines in THE presence of recombinant sIFNAR2. PMA / ionomYcin A was used as positive control to stimulate cytokine production and UNstimulated cells were used as negative control.”
Also, word order is often inverted, e.g. line 600 “the soluble forms of cytokine receptors biologically active”, ”Figure Supplementary” etc.
Please use correct names for cytokines and use consistently throughout the manuscript (text and figures): GM-CSF (not GM-CFS), IFN-alpha or IFN-α (not IFNa), idem for IFN-gamma, TNF-alpha, also IL-10 (not IL10) etc.
Reviewer 2 Report
In this paper, the authors explored the use of soluble IFNAR2 as a potential therapeutic for inflammatory diseases. Here they cloned the alternatively splice sIFNAR2 gene in order to produce sIFNAR2 protein. sIFNAR2 protein is a naturally occurring protein that modulates Type I interferon activity. In order to study the modulatory activity of sIFNAR2 as it relates to Type I interferon activity, the authors used PBMC from 19 different individuals and then stimulated these cells in the presence of PMA/ionomycin with or without their preparations of sIFNAR2 protein. Cytokine production was followed in Luminex assays, as well as Intracellular cytokine staining with FACS analysis. Another property if IFN-beta is it anti-proliferative effect and the activity of sIFNAR2 in this was examined in percent confluence assays. A540 cells were challenged with EMCV virus and the antiviral effect of sIFNAR2 was evaluated. The results show that sIFNAR2 decreased most cytokine responses, but not all. Also the sIFNAR2 had a very potent antiviral effect nearly eliminating EMCV cytopathology.
The central hypothesis is innovative and the data for the most part are solid. There are several concerns on some of the data that need to be addressed.
The effect on sIFNAR2 on the intracellular IL-17 was significant but very slight. One wonders if this is biologically meaningful. Also these values do not match with the secreted IL-17 data where the sIFNAR2 clearly eliminated the IL-17 response. Please recognize this in the text or eliminate the data on intracellular staining, which is redundant and a little contradictory. Also, it is not clear as to the significance of the the antiproliferative effect of sIFNAR2 on Neuro2a cells. Clearly proliferation remained robust in the presence of either IFN-beta or sIFNAR2. The recommendation is that the authors remove these data. The antiproliferative effect is not clear using the percent confluence assay. Recommendation is to use CCK8 assays.
Reviewer 3 Report
In this manuscript, the authors test the effects of sIFNAR2 on human cells, showing that it has some effects similar to that of IFNB, although it does not appear to activate the same signalling pathways as IFNB. They show that when sIFNAR2 was added to human cells, it suppressed cytokine production and proliferation, and reduced he cytopathic effect of viral infection.
There are some concerns regarding the controls used for this study, and the preparation of reagents, which prevent firm conclusions from being drawn from this data.
Firstly, the sIFNAR2 used in this study was His-purified from transgenic bacteria. E coli expression always results in some contamination of purified protein with bacterial products, in the absence of further purification methods, with (for instance) LPS being co-purified with protein. The authors do not state whether levels of LPS were measured in their protein preparation, however a previous publication of theirs (Suardiaz, 2016, Neuropharmacology) states that levels of endotoxin "was less than 1 EU/ug, which was considered to be acceptable for animal use". Although this level of endotoxin contamination will not kill a mouse, it can have significant effects on the immune response. At the concentration of protein used in this study (~30ug/ml) that is equivalent of up to 3 ng/ml of LPS being added to wells, which could be enough to inhibit viral infectivity (Chen, J Virology, 1999, 73:2650). This may not be a problem but it must be excluded before conclusions can be drawn from these results. Ideally, a control protein should be expressed in the same system, as this should have similar contaminants, and the same tags etc. Alternatively, the protein could be treated with proteases to cleave it or heat-treatment to denature it, and show abrogation of activity. Especially in this system, where the mechanism of action is unclear, an investigation of possible contaminants is very important. Potential contaminants could explain the large batch to batch variation of the effect, as Figure 4C shows a near 100% suppression of cytopathic effect with some batches of protein at 30 ug/ml, but double the amount having to be used for a similar effect in Fig 4D.
Controls tested included "elution buffer" - Fig 4A. As the protein had been dialysed out of elution buffer, this does not seem an appropriate control. What is the elution buffer used here? At the "B2" concentration of buffer used, there did appear to be an increase in viability (~30% to ~50%), but this is not mentioned. Use of albumin as an irrelevant protein control (Fig 4B) does not control for contaminants and charged His tags, as mentioned above.
High concentrations (>10 ug/ml) of sIFNAR2 were used to achieve the effects seen - would these concentrations ever be reached in vivo?
Line 455: "B18" is introduced here, without saying what it is - an antibody, or a signalling inhibitor? A reference should be cited.
Fig 5D: RNAseq is used to compare gene transcription after either IFNB or sIFNAR2. Although the lack of IFNB-like responses is noted, the changes seen with sIFNAR2 are not discussed.
Lines 515-517 should be deleted.
Line 573: replace "quantity" with "quality"
Line 595: "No synergistic effects were observed..." As a likely saturating concentration of sIFNAR2 was used in these experiments, it would be unlikely to see any synergy - a sub-saturating concentration of both sIFNAR2 and IFNB would have to be used to make this statement, or the statement should be removed.
Line 608: was "sIFNAR2" meant here instead of "IFNB"?
Line 619: Replace "RNA sec" with "RNA-seq"